# Prevalence of hypertension and its associated factors among government employees in Doti district of Nepal

Ramesh Malashi[1], Sunita Sharma[1]*, Srijana Adhikari[1], Chitra Raj Sharma[2], Arun Kumar Joshi[3], Buna Bhandari[1,4]

**1** Central Department of Public Health, Institute of Medicine, Tribhuvan University, Maharajgunj, Kathmandu, Nepal, **2** National Academy of Medical Sciences, Bir Hospital, Kathmandu, Nepal, **3** Chitwan Medical College, Chitwan, Nepal, **4** Department of Global Health and Population, Harvard T.H Chan School of Public Health, United States of America

* springdale2424@gmail.com

## Abstract

### Introduction

Hypertension is a significant risk factor for cardiovascular diseases (CVDs), which remains the leading causes of morbidity and mortality globally, with a disproportionate impact on low and middle income countries. While hypertension is prevalent across various populations, government employees are particularly susceptible due to high stress levels, sedentary lifestyles, and work-related pressures. Therefore, this study was undertaken to assess the prevalence of hypertension and its associated risk factors among government employees in the Doti district of Nepal.

### Methods

A cross-sectional study was carried out among 195 government employees in Dipayal Silgadhi Municipality of Doti district of Sudurpashchim province of Nepal. The data was collected through face-to-face interviews using Simple Random Sampling (SRS) technique and analysed using SPSS v25. The structured questionnaire adopted from the WHO STEPS survey tool was used for data collection. Bivariate and multivariate logistic regression model was used to assess the factors associated with hypertension.

### Results

The prevalence of hypertension among government employee was 36.4%%±5.6%. Participants with age group 30–40 years [Adjusted Odd's Ratio (AOR) 14.4, 95% Confidence Interval (CI) (1.6, 127.7)], 40–50 years [AOR 13.7, 95% CI (1.04, 180.3)] and work experience (20–30 years) [AOR 6.67, 95% CI(1.23, 35.9), and drinking alcohol [AOR 0.35, 95% CI (0.17, 0.72)] were found to be statistically significant with hypertension.

**Data availability statement:** All relevant data are within the manuscriptand its Supporting Information files.

**Funding:** The author(s) received no specific funding for this work.

**Competing interests:** The authors have declared that no competing interests exist.

## Conclusion

The study revealed the high prevalence of hypertension among government employees; significantly associated with risk factors like age group 30–50 years, work experience and alcohol consumption, indicating an alarming public health concern. These results highlight the pressing need for focused interventions to reduce the risk of hypertension and its related problems among government employees, such as lifestyle changes, workplace health programs, and routine health screenings.

## Introduction

Hypertension is a leading modifiable risk factor for cardiovascular diseases (CVDs) and premature mortality worldwide. Between 1990 and 2019, global deaths from CVDs increased steadily from 12.1 million to 18.6 million respectively [1]. The prevalence of hypertension among adults is higher in Low Middle Income Countries (LMICs) (31.5%, 1.04 billion people) than in High-Income Countries (28.5%, 349 million people) [2]. The rising prevalence of hypertension, coupled with low medication adherence, poor blood pressure control, and limited knowledge about the condition, presents significant challenges, particularly in low- and middle-income countries (LMICs) [2].

Hypertension has emerged as a major global health challenge, with the 2019 Global Burden of Disease (GBD) study highlighting it as a leading contributor to disability-adjusted life years (DALYs), largely because of its connection with serious conditions like ischemic heart disease, stroke, and chronic kidney disease [3]. In Nepal, a systematic review and meta-analysis revealed that 28.52% of individuals were affected by hypertension, with key risk factors including body mass index (BMI), educational status, tobacco use, alcohol consumption, and physical activity [4]. Similarly, a study conducted in Pokhara found a staggering 66.2% age-adjusted prevalence of hypertension, with alcohol use, obesity, and a family history of hypertension being significant contributing factors [5]. These statistics paint a stark picture of the growing hypertension crisis and underscore the need for urgent action to address this silent yet dangerous epidemic.

According to various epidemiological studies, stress, obesity, and a sedentary lifestyle are important risk factors for hypertension [6,7]. Government employees are the working population who are often exposed to sedentary lifestyles, prolonged work-related stress, irregular eating habits, and limited physical activity due to desk-bound job responsibilities. These occupational risk factors may increase their vulnerability to developing non-communicable diseases such as hypertension [6]. A study conducted among civil servants in South Africa reported the prevalence of hypertension was 24.5% among 546 participants, concluding that one in four civil servants is hypertensive [8]. Similarly, study conducted among bankers of Kathmandu reported the prevalence of hypertension of 11.3% and its significant association with gender, diabetes and physical inactivity [9].

While several studies have focused on the general population, there remains a significant knowledge gap regarding the prevalence of hypertension and its associated factors within specific groups, like data from government officials which forms the rationale behind this study. Doti district, one of the underprivileged rural and hilly areas of Nepal has insufficient health data, especially when it comes to non-communicable diseases like hypertension. Studying prevalence of hypertension among government employees in a rural setting like Doti district not only closes a significant evidence gap but also contributes to the development of targeted strategies for the prevention and management of hypertension in underprivileged areas of Nepal. Therefore, the primary objective of this study was to assess the prevalence of hypertension and its associated factors among government employees in Dipayal Silgadhi Municipality, Doti district, Nepal.

## Materials and methods

### Study design and setting

An analytical cross-sectional study was conducted at Dipayal Silgadhi Municipality of Doti District, one of the hilly districts of Far Western Province, Nepal. The district covers an area of 2025 square kilometers. The district has a population of 205683 and households of 36806 as per census 2022 AD [10].

### Study population

The study population consisted of Government employees working in civil service, health service, and miscellaneous service within the administrative area of Dipayal Silgadhi Municipality of Doti district. Miscellaneous service includes other gazetted-level government employees falling outside administration or health like audit, foreign affairs, statistics, engineering, agricultural, and other services. Currently employed government staffs that were present during the data collection period were included in the study, while those who were on leave or absent were excluded.

### Sample size

The total calculated sample size of 195 was determined using the statistical formula by Cochran for a finite population. The expected proportion (p) in population was based on National Demographic Health Survey (NDHS) of Nepal = 0.20, i.e., prevalence of hypertension among adult population [11] with a 95% confidence interval and a 5% absolute margin of error. Since the study population was limited to government employees in Doti district, the total number of government employees in Doti (N = 625).

$$\text{Sample size (n)} = n_0 / \{1 + (n_0 - 1)/N\}$$

Hence,

$$n_0 = \frac{(1.96)^{2*}0.2*0.8}{(0.05)^2}$$

$$= 245.86 \approx 246$$

$$= 245.86 \approx 246$$

Now,

$$\text{required sample (n)} = n_0 / \left\{ 1 + (n_0 - 1)N \right\}$$

$$= 246/\{1 + (246 - 1)/625\}$$

$$= 176.7 \approx 177$$

Considering 10% non- response rate, total required sample size = 177 + 18 = 195

## Sampling technique

The probability sampling (simple random sampling) method was used in the study. A list of government offices was collected from the Municipality office of Dipayal Silgadhi Municipality and the District Administration Office (DAO) of Doti district. Then, a list of government employees was obtained from respective offices, and a sampling frame was prepared. The total number of Government employees within the Dipayal Silgadhi Municipality was 625. The required sample was selected by SRS technique using lottery method. The individual sample was informed about the objective, process, and written consent for the study was taken before data collection.

## Data collection tools and technique

Data was collected from 5th December 2022–4th January 2023. It was collected through Face-to-face interviews using a standard structured questionnaire with the help of two trained enumerators (Health Assistants) after one day of orientation. Questionnaire was adopted from the STEP survey tools of WHO, which have been validated in Nepali by Nepal Health Research Council (NHRC) [12]. Total 210 participants were approached for data collection, among them 195 participants agreed to participate in the study with the response rate of 93%.

## Blood pressure measurement

The Blood pressure of the participants was measured as per the guidelines provided by Centers for Disease Control and Prevention (CDC) 2021 [13] and measurement of blood pressure was done by using the Omran HBP-1300 upper arm automatic blood pressure monitor. As per the CDC guidelines, data was collected only after confirming that the participants had their last meal intake an hour before. Participants were further asked about their last smoking and alcohol consumption as it may also alter the blood pressure measurement. None of the participants reported non-consumption of smoking or alcohol before 30 minutes at the time of data collection.

Before the Blood Pressure measurement, participants were asked to sit comfortably in sitting position and then blood pressure was measured using Omran HBP-1300 upper arm automatic blood pressure monitor in the right arm. Three BP measurements were taken at three-minute intervals after participants rested for at least ten minutes. The average of the last two measurements was included in the study. Individuals were defined as hypertensive if their Systolic Blood Pressure (SBP) was 140 mmHg or higher and/or if their Diastolic Blood Pressure (DBP) was 90 mmHg or higher or if they were taking antihypertensive medication [14].

Pre-hypertension: SBP = 120–139 mmHg and/or DBP = 80–89 mmHg Stage 1 Hypertension: SBP = 140–159 mmHg and/or DBP = 90–99 mmHg Stage 2 Hypertension: SBP ≥ 160 mmHg and/or DBP ≥ 100 mmHg [14]. Data collection began on 5th December 2022 and was completed on 4th Jan 2023.

## Independent Variables

Socio-demographic variables: Age, Sex, Marital status, Ethnicity, Religion, Family history of hypertension
Socio-economic variables: Education, Service, Income (personal)
Behavioural variables: Diet (Fruit, vegetables, salt), Physical activity, Tobacco use, Alcohol use

## Validity and Reliability of the tools

The validated questionnaire used in STEP Survey (WHO) was used in the study. Content validity was ensured by developing the questionnaire according to the objectives of the study and study variables. Frequent consultation was done with the subject experts and research supervisors while developing the tools. Pretesting of questionnaire was done among 10% of similar sample and ambiguous words with vague terms were modified post pre-testing. For consistent and reliable information, one day orientation was provided to the enumerators for data collection and supervision was sought throughout the data collection process.

## Data management and analysis

The collected data were checked manually to minimize the possible errors and the data were systematically coded and entered in Epi-Data 3.1. The entered data were exported to IBM-SPSS version 25 for further analysis.

Using univariate analysis, frequency and percentages of socio-demographic, behavioral and dependent variables were presented in a frequency table. Bivariate analysis was done to assess the factors associated with hypertension and p-values at a 95% level of confidence were reported. The multivariate analysis was done between independent and dependent variables which were found to be statistically significant in bivariate analysis. Multicollinearity was assessed using Variance Inflation Factor (VIF) diagnostics with a cutoff of 10 and the goodness of fit of the logistic regression model was assessed using the Hosmer–Lemeshow test.

## Ethical considerations

Ethical approval was obtained from Institutional Review Committee (IRC), Institute of Medicine, Tribhuvan University [Ref. no.258(6–11)E22079/080]. Before data collection, approval was obtained from respective government offices. Written informed consent, after explanation about the study, was obtained from the study participants.

## Results

Total 210 participants were approached for data collection: among them 195 participants agreed to participate in the study with the response rate of 93%. The mean ±SD age of participants was 38.76 ± 11.32 years and the age range of 20 years to 60 years. There were 71.8% male participants, followed by females (28.2%). 89.8% of participants were Brahmin/Chhetri followed by Dalit (5.1%) and Janajati (5.1%). About one third (33.3%) of them had bachelor and higher level of education and most of them (85.6%) were married. In addition, most of the participants (71.3%) were doing civil service, followed by health service (18.5%) and miscellaneous services (10.3%), as presented in Table 1.

The prevalence of hypertension among government employee was found to be 36.4% ± 5.6% (95% confidence, with finite population correction) as presented in Table 2. The margin of error for sample size (n = 195) and total number of government employees in Doti (N = 625) with 95% Confidence Interval has been calculated as follows,

**Step 1: Standard Error (without FPC)**

$$SE = \sqrt{\frac{p(1-p)}{n}} = \sqrt{\frac{0.364(1-0.364)}{195}} = \sqrt{0.0011872} \approx 0.03447$$

**Step 2: Finite Population Correction (FPC)**

$$FPC = \sqrt{\frac{N-n}{N-1}} = \sqrt{\frac{625-195}{624}} = \sqrt{\frac{430}{624}} = \sqrt{\{0.688\}} \approx 0.8294$$

**Table 1. Socio-demographic characteristics of study participants (n = 195).**

| Variables | Number | Percentage |
|---|---|---|
| **Age (in years)** | | |
| Mean age 38.76 years, SD = 11.32 years | | |
| 20–30 | 63 | 32.3 |
| 30–40 | 56 | 28.7 |
| 40–50 | 30 | 15.4 |
| 50–60 | 46 | 23.6 |
| **Sex** | | |
| Male | 140 | 71.8 |
| Female | 55 | 28.2 |
| **Ethnicity** | | |
| Dalit | 10 | 5.1 |
| Janajati | 10 | 5.1 |
| Brahmin/Chhetri | 175 | 89.8 |
| **Education** | | |
| Less than primary level | 27 | 13.8 |
| Primary level completed | 25 | 12.8 |
| Secondary level completed | 52 | 26.8 |
| Intermediate level | 26 | 13.3 |
| Bachelor and higher level | 65 | 33.3 |
| **Marital Status** | | |
| Married | 167 | 85.6 |
| Unmarried | 28 | 14.4 |
| **Service type** | | |
| Civil service | 139 | 71.3 |
| Health service | 36 | 18.4 |
| Miscellaneous service | 20 | 10.3 |

**Table 2. Prevalence of Hypertension (n = 195).**

| Characteristics | Number | Percentage |
|---|---|---|
| Non-Hypertensive (<140/90 mm of Hg) | 124 | 63.6 |
| Hypertensive (≥140/90 mm of Hg) | 71 | 36.4 |

## Step 3: Corrected Margin of Error (MoE)

$$\text{MoE} = z \times \text{SE} \times \text{FPC} = 1.96 \times 0.03447 \times 0.8294 = 1.96 \times 0.02859 \approx 0.05604$$

$$\text{Margin of Error} = 5.6\%$$

## Association of socio-demographic variables and hypertension

The table below shows the multivariate analysis of socio-demographic variables with hypertension. Age group of the participants and working experience were found statistically significant with hypertension in multivariate analysis. Participants in age group 30–40 years [AOR = 14.4 CI 1.6, 127.7] and 40–50 years [AOR = 13.7, CI (1.04, 180.3)] were found to be

statistically significant. The result also revealed that the participants who had more than 20 years of working experience were statistically significant with hypertension as presented in Table 3.

While applying the multivariate analysis, the Hosmer and Lemeshow test was performed to test the goodness of fit. This test found the model to be fit (p = 0.345), which was greater than the cut-off point of value (p = 0.05). The coefficient of determinant (Nagelkerke R square) for the equation was 0.445, which explained that the independent variables explained about 45% of the change in the dependent variable.

### Association of behavioral factors with hypertension

Upon multivariate regression analysis, ever drinker was statistically significant with hypertension [AOR 0.35, 95% CI (0.17, 0.72)] as presented in Table 4.

The Hosmer and Lemeshow test was used to test the goodness of fit, which found the model to be fit (p = 0.820). The coefficient of determinant (Nagelkerke R square) for the equation was 0.091, which explained that 9.1% of the change in the dependent variable was explained by the independent variables.

## Discussion

The prevalence of hypertension among government employees in this study was found to be 36.4% (95% CI: 30.8%–42.0%) and the factors associated with hypertension were age group 30–50 years, work experience, and alcohol consumption.

**Table 3. Association of socio-demographic variables with hypertension (n = 195).**

| | COR | p-value | AOR | p-value |
|---|---|---|---|---|
| Variables | CI (95%) | | CI (95%) | |
| **Age group** | | | | |
| 20–30 | Ref | | | |
| 30–40 | 0.02 (0.01, 0.09) | **0.001*** | 14.4 (1.6, 127.7) | **0.016*** |
| 40–50 | 0.25 (0.11, 0.57) | **0.001*** | 13.7 (1.04, 180.3) | **0.046*** |
| 50–60 | 0.73 (0.28, 1.89) | 0.511 | 11.6 (0.73, 6.5) | 0.082 |
| **Sex** | | | | |
| Female | Ref | | | |
| Male | 5.77 (2.4, 13.64) | **0.001*** | 2.25 (0.79, 6.5) | 0.130 |
| **Education** | | | | |
| Less than primary level | Ref | | | |
| Primary level | 0.62 (0.21, 1.88) | 0.406 | 0.88 (0.23, 3.32) | 0.860 |
| Secondary level | 0.38 (0.15, 1.01) | 0.052 | 0.82 (0.25, 2.62) | 0.746 |
| Intermediate level | 0.10 (0.03, 0.43) | **0.002*** | 0.36 (0.06, 2.02) | 0.249 |
| Bachelor or higher | 0.5 (0.20, 1.24) | 0.135 | 1.31 (0.39, 4.37) | 0.654 |
| **Marital status** | | | | |
| Unmarried | Ref | | | |
| Married | 5.72 (1.66, 19.71) | **0.006*** | 0.28 (0.02, 2.94) | 0.292 |
| **Work experience (in years)** | | | | |
| Less than 10 | Ref | | | |
| 10–20 | 3.86 (1.47, 10.12) | **0.006*** | 1.83 (0.55, 6.03) | 0.319 |
| 20–30 | 13.7 (5.7, 32.8) | **0.001*** | 6.67 (1.23, 35.9) | **0.027*** |
| 30–40 | 12.3 (4.4, 34.5) | **0.001*** | 6.09 (0.81, 45.9) | 0.080 |

*Statistically significant association with hypertension.

**Table 4. Association of behavioral factors with hypertension (n = 195).**

| Variables | Crude OR (CI 95%) | p-value | Adjusted OR (CI 95%) | p-value |
|---|---|---|---|---|
| **Ever smoking tobacco** | | | | |
| No | Ref | | | |
| Yes | 2.16 (1.19, 3.93) | **0.011*** | 0.77 (0.38, 1.57) | 0.486 |
| **Ever drinker** | | | | |
| No | Ref | | | |
| Yes | 3.18 (1.72, 5.91) | **0.01*** | 0.35 (0.17, 0.72) | **0.005*** |
| **Frequency of Smokeless tobacco consumption** | | | | |
| Less than 5 | Ref | | | |
| 5–10 | 1.01 (0.26, 3.92) | 0.985 | 0.66(0.12, 3.64) | 0.640 |
| 10–15 | 3.25 (0.7, 15.07) | 0.132 | 0.79 (0.10, 5.97) | 0.820 |
| More than 15 | 5.57 (1.12, 27.5) | **0.035*** | 2.31 (0.34, 15.7) | 0.391 |
| **Frequency of drinking** | | | | |
| Occasionally | Ref | | | |
| Daily | 1.96 (0.53, 7.20) | 0.309 | 1.32 (0.24, 7.29) | 0.745 |
| Weekly | 4.90 (1.19, 20.1) | **0.027*** | 1.28 (0.20, 7.93) | 0.787 |
| Once a month | 3.27 (3.27, 19.3) | 0.191 | 0.62 (0.04, 9.02) | 0.731 |

*Statistically significant association with hypertension.

Globally, hypertension affects approximately one-third of the adults, about 31.1% as estimated by a global systematic review, reflecting a growing burden of cardiovascular disease worldwide [2]. The prevalence of hypertension is disproportionately higher in low- and middle-income countries (LMICs) at 31.5%, compared to 28.5% in high-income countries [2].

Within the occupational groups, prevalence of hypertension tends to vary by region and lifestyle related factors. Studies among Ethiopian civil servants reported rates between 24.5% [8] and 27.3%, while Chinese working adults showed similar figures at 28.1% [15], both lower than the findings of our study. However, the army veterans of the India Gorkha regiment reported higher rate of prevalence of 66.2% [5] and in Bangladesh, government employees demonstrated the prevalence of hypertension as 38.3% [16]. This disparity in the findings may be due to differences in occupational stress, dietary patterns, physical activity, healthcare access, and substance use behaviors across countries. Additionally, methodological differences such as sample size, measurement criteria, or population age distribution could have contributed to variations in reported prevalence. The community-based studies conducted in Kathmandu reported the prevalence of hypertension as 32.5% which is similar to our study's findings [17].

This study further identified age (30–50 years), work experience, and alcohol consumption as significant factors associated with hypertension. These align with the findings from both national and international studies. The increasing age (age group more than 35 years) and alcohol use were significant in Bangladeshi government employees [17]. However, adjusted odds ratios (AORs), particularly for age groups 30–40 (AOR: 14.4; 95% CI: 1.6–127.7) and 40–50 (AOR: 13.7; 95% CI: 1.04–180.3), have very wide confidence intervals, suggesting a high degree of uncertainty around the estimates. This is likely due to small subgroup sizes or sparse data.

Interestingly, alcohol consumption in our study showed an inverse association (AOR:0.35; 95% CI: 0.17–0.72), potentially reflecting reverse causation, reporting bias, or unmeasured confounding. It is possible that this observed association is influenced by residual confounding or reverse causation, whereby individuals with existing health conditions may reduce or abstain from alcohol use. Moreover, potential underreporting or misclassification of alcohol intake cannot be ruled out. This is a notable limitation of the study and underscores the need for longitudinal research to further explore this relationship.

A cross-sectional study conducted in Kathmandu reported that over three-quarters (78.4%) of participants engaged in sufficient physical activity [17], a figure notably higher than that observed in the present study. This discrepancy could be attributed to the nature of government office work, where employees often spend prolonged hours in sedentary desk jobs, limiting opportunities for regular physical movement and contributing to reduced activity levels.

One of the studies conducted among Bhutanese adults reported that 47% of the study population were found to be currently drinking alcohol [18], which is higher than this study (33.3%). But the findings of this study is higher than that conducted in Kathmandu (27%) [17].Similarly, the current smokers found in this study were higher than that was found by STEPS survey, 2019 (29%) [19].This data showed the increasing trend of risk factors of hypertension among Government employees.

In this study, only 29.7% of participants consumed sufficient number of fruits and vegetables. In contrast, the STEPS survey 2019 reported that 96.7% consumed fewer than five servings of fruits and vegetable per day, indicating insufficient intake of fruit and vegetables consumption [19]. This contrast could be due to the difference in study participants, their education level and purchasing capacity. This study population were employed with regular earning capacity.

According to the European society of cardiology, there is a well-established association between high sodium consumption and hypertension where it clearly highlighted that processed and canned foods often contain high level of sodium which can increase in blood pressure [20]. While in this study, around 11.8% of participants added salt always or often during taking meal, which is higher than that was found in STEPS survey, 2019(5.6%). Additionally, the STEPS survey also reported that 19.5% of study population consumed processed food always or often which is slightly higher than that was found in this study (14.4%) [19].

In conclusion, our findings reveal a concerning elevation in hypertension among government employees which is a notable concern. Interventions tailored to workplace environments such as encompassing routine screenings, healthy diet and physical activity promotion, and alcohol education [21] are critically needed to address this emerging public health challenge.

## Limitations of the study

The study adopted a cross-sectional design, which may preclude the establishment of causality with the identified risk factors. Additionally, measurements of fruit and vegetable consumption, salt intake, and behavioral factors such as tobacco use, alcohol consumption, and physical activity were based on self-reports rather than objective measurements, which may introduce reporting and recall biases. However, the researchers made efforts to ensure accurate responses through careful probing.

The response rate of our study was 93%. Since not all approached participants agreed to participate, we have also noted this as a limitation.

Alcohol consumption showed an inverse association with hypertension; however, this should be interpreted with caution due to the cross-sectional design, which limits causal inference. The association may be influenced by residual confounding, reverse causation, or underreporting. The Hosmer-Lemeshow test (p = 0.345) suggested a good model fit, though its accuracy is limited in small samples (n = 195). Despite these constraints, the study provides valuable insights into hypertension and its risk factors among government employees in Nepal.

## Conclusion

This study revealed that the prevalence of hypertension among government employees as ranging from 30.8% to 42.0% which is a significant concern. The identified risk factors, including increasing age, work experience and alcohol consumption, call for targeted interventions to mitigate hypertension-related complications within this population. This study provides crucial evidence that can inform the development of evidence-based policies, plans, and programs aimed at reducing modifiable occupational risk factors of hypertension among high-risk groups. The appropriate interventions

including lifestyle modification and workplace health programs should be planned and carried out to address the high prevalence of hypertension among government employees.

## Supporting information

**S1 Table. Behavioral characteristics of the participants and its association with Hypertension.**
(DOCX)

**S1 File. Operational Definitions.**
(DOCX)

## Acknowledgments

We would like to acknowledge Prof Amod Kumar Poudyal, Mr. Susan Man Shrestha and faculties of Department of Community Medicine, TU, IOM for their support for conducting this study. Similarly, we are grateful to Chiefs of Government Offices within Dipayal Silgadhi Municipality for their support for data collection and also appreciate Mr. Rabindra Bhandari for his guidance in data analysis of this study.

## Author contributions

**Conceptualization:** Ramesh Malashi, Sunita Sharma, Chitra Raj Sharma, Buna Bhandari.

**Data curation:** Ramesh Malashi, Sunita Sharma.

**Formal analysis:** Ramesh Malashi, Chitra Raj Sharma.

**Investigation:** Ramesh Malashi, Buna Bhandari.

**Methodology:** Ramesh Malashi, Chitra Raj Sharma, Buna Bhandari.

**Project administration:** Arun Kumar Joshi.

**Resources:** Srijana Adhikari.

**Software:** Ramesh Malashi.

**Supervision:** Sunita Sharma, Chitra Raj Sharma, Buna Bhandari.

**Validation:** Sunita Sharma, Chitra Raj Sharma, Buna Bhandari.

**Visualization:** Arun Kumar Joshi.

**Writing – original draft:** Ramesh Malashi, Sunita Sharma.

**Writing – review & editing:** Sunita Sharma, Srijana Adhikari, Chitra Raj Sharma, Arun Kumar Joshi, Buna Bhandari.

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
