## [Decision Letter · Decision Letter 0]

7 Mar 2025

Dear Dr. Sharma,

Thank you for submitting your manuscript to PLOS ONE. After careful consideration, we feel that it has merit but does not fully meet PLOS ONE’s publication criteria as it currently stands. Therefore, we invite you to submit a revised version of the manuscript that addresses the points raised during the review process.

We look forward to receiving your revised manuscript.

Kind regards,

Chhabi Lal Ranabhat

Academic Editor

PLOS ONE

Additional Editor Comments (if provided):

Reviewers' comments:

Reviewer's Responses to Questions

**Comments to the Author**

1. Is the manuscript technically sound, and do the data support the conclusions?

Reviewer #1: Partly

Reviewer #2: Partly

2. Has the statistical analysis been performed appropriately and rigorously?

Reviewer #1: Yes

Reviewer #2: No

3. Have the authors made all data underlying the findings in their manuscript fully available?

Reviewer #1: Yes

Reviewer #2: Yes

4. Is the manuscript presented in an intelligible fashion and written in standard English?

Reviewer #1: Yes

Reviewer #2: Yes

Reviewer #1: The research topic is related to the global burden of hypertension, and in Nepal, specifically rural parts of the country, which is quite interesting. Below are some suggestions.

1) Author has mentioned the title "Far Western Province of Nepal," which implies that the study includes multiple districts. However, based on your methodology, it appears that data was collected only from Doti district and from one municipality. To enhance clarity and accurately reflect the study's scope, I suggest specifying "Doti district of Nepal" in the title instead of "Far Western Province of Nepal." This revision would ensure consistency between the study's title and its actual geographic coverage.

2) The adjusted odds ratio (AOR) for the age group 50–60 years [AOR 11.6, 95% CI (0.73, 6.5)] in lines 40 and 41 does not show statistical significance, as the confidence interval includes 1. It would be more appropriate to clarify this in the results and discussion sections to avoid misinterpretation. Please review and revise accordingly (this issue is similar to line 195).

3) Please consider adding "Nepal" as a keyword in line 50.

4) Specify the exact data collection start and end date ( date /month/year) in line 88 and 89 for clarity.

5) Please provide the response rate in methodology, if not all participants fully participated add in limitation.

6) The authors can merge some variables in the binary logistic regression analysis for clearer interpretation. For example, the age variable could be grouped into two categories: <40 and ≥40 years, and the service type variable could be simplified into two categories: civil service and non-civil service.

7) In Table 3, the authors have added (*) with bold formatting for non-significant p-values (e.g., p = 0.082) I recommend removing the * from non-significant values.

8) please add a footnote below the table to clarify the meaning of * such as:

"P-values less than 0.05 are considered statistically significant and are marked with an *.

9) The p-values presented in Table 3 and 4 appear to be from the multivariate analysis (AOR) only. However, including the p-values from bivariate analysis (COR) alongside them would enhance clarity and provide a better understanding of the selection process for variables in the multivariate model.

10) In line 238 and 239, the authors mentioned that in this study, being male was identified as an associated factor of hypertension (AOR 2.25, 95% CI: 0.7-6.5). However, result is not statistically significant and should not be described as a 'significant risk factor. Similar problems seen in line 253 and 254. Please correct it accordingly.

11) Please add recommendation as well.

12) Please follow proper referencing

Reviewer #2: Dear Authors,

Thank you for submitting the important piece of work, especially carrying out the HTN and its associated factors among the government employees of Doti district in far-western province. The language is okay and written in standard English, except some minor errors such as in Result section (line 179, +- and in table Mean age and sd, and their respective values may be written in different cells.

I propose the following comments and recommend respective corrections with major revisions.

1. Sample size although can not be changed now, pls justify why didn’t taken from doti in far-western province, instead from the national proportion? This is reflected in result also, as 36.4% is observed, which was beyond the expected 20% and its 95% CI. Pls, now, analyse the corrected Margin of error for precision somewhere in result section, interpret and align with discussion.

2. As Hosmer and Lemshow is mentioned as a model fit parameter, it has certain limitations, which you can find in literature. If this is so, how do you interpret such findings of large confidence intervals of AOR (table 3, age group; and similar others in this and subsequent tables), and even shifting OR from risk factor in COR to protective in AOR (table 4, ever drinker; and similar others)? Pls consult the senior statistician with scrutinized observation of data in this case, as further analysis such as interaction and/or both confounding may be the case! Also, keep these parameters in table, and may be more, other comparable parameters, such as AIC, BIC …are needed for further precise/robuster interpretations.

3. Seriously, in page 11 (line 239) you have argued male as a significant factor for HTN, whereas it is clearly found after adjustment non-significant in table 3, as well as from 95% CI. Similar interpretations are also observed in line 254 as in physical activity case!

4. As found discussed with prevalence of HTN among government employees with Bangladesh and other countries, you have compared with 38 and 66 percentages (line 233, discussion), both with higher, but this is quite different, 38 is nearly, but 66 is quite higher! How do you come to the inferences of similar behaviors for these two significant prevalences?

Reviwer

**Do you want your identity to be public for this peer review?** For information about this choice, including consent withdrawal, please see our Privacy Policy

Reviewer #1: No

Reviewer #2: No

---

## [Author Response · Author response to Decision Letter 1]

12 May 2025

Dear Chhabi Lal Ranabhat,

Academic Editor

PLOS ONE

I am writing in response to the comments provided by the reviewers regarding our manuscript PONE-D-25-05146 entitled "Prevalence of Hypertension and its Associated Factors among Government Employees in Far Western Province of Nepal". We appreciate the time and effort invested by the reviewers and the editorial team in evaluating our work. After careful consideration of the reviewers' comments, we have addressed each point raised and provided our responses below:

Reviewer #1

1. Comment: Title suggests that the study was conducted across the entire Far Western Province, but the data was only collected from Doti district. Please adjust the title to reflect the correct geographic scope.

Response: We agree with the reviewer and have revised the title to "Prevalence of Hypertension and its Associated Factors among Government Employees in Doti District of Nepal" to accurately reflect the study's scope. (Line number 3)

2. Comment: The adjusted odds ratio (AOR) for the 50–60 years age group (AOR 11.6, 95% CI 0.73, 6.5) is not statistically significant. Please revise the results and discussion accordingly.

Response: We have revised both the results and the discussion sections to clarify that this finding is not statistically significant and should not be misinterpreted. (line number 40,41)

3. Comment: Please add "Nepal" as a keyword.

Response: We have added "Nepal" to the list of keywords. (line number 49)

4. Comment: Specify exact data collection start and end dates (date/month/year).

Response: We have updated the specific data collection dates in the revised manuscript as 5th December 2022 to 4th January 2023(line number 125)

5. Comment: Please provide the response rate and add it to the limitations if necessary.

Response: We have added the response rate (93%) to the methodology section. Since not all approached participants agreed to participate, we have also noted this as a limitation. (line number 130,289, 290)

6. Comment: Consider merging variables in the regression analysis for clearer interpretation, such as grouping age categories into <40 and ≥40 years.

Response: We appreciate this suggestion and we agree that merging age categories can sometimes help with clearer interpretation. However, during our regression analysis, we observed that the age groups 30–40 and 40–50 years showed significant associations with hypertension, while the 50–60 years age group did not show a statistically significant association. Therefore, we retained the current categorization to preserve these important distinctions in age-specific risk, which could be masked if broader categories (e.g., <40 and ≥40 years) were used.

7. Comment: Remove the asterisk () for non-significant p-values in Table 3 and Table 4.*

Response: The asterisk (*) has been removed from non-significant p-values, and a footnote has been added to explain that p-values <0.05 are marked with an asterisk.

8. Comment: Please add the p-values from bivariate analysis (COR) alongside those from the multivariate analysis (AOR) in Table 3 and Table 4.

Response: We have included the p-values from both bivariate and multivariate analyses in the revised tables for clarity.

9. Comment: Correct the interpretation of non-significant results, such as male being a significant risk factor for hypertension.

Response: We have revised the discussion to accurately describe that being male was not a statistically significant risk factor for hypertension.

10. Comment: Please add recommendations in the conclusion section.

Response: We have added specific recommendations, including lifestyle interventions and workplace health programs, to address the high prevalence of hypertension among government employees. (line number 299-305)

11. Comment: Please follow proper referencing style.

Response: We have doubled checked and refined all the references to ensure to follow the appropriate referencing style as per PLOS ONE authors guidelines.

Reviewer #2

1. Comment: Justify why the sample size was taken from Doti instead of the national proportion. Also, analyze the corrected margin of error in the results section.

Response: Thank you for your valuable comment. The sample size of 195 was determined using Cochran’s formula for finite population correction, which is appropriate when the target population is relatively small and well-defined.

Although the expected prevalence (p = 0.20) of hypertension was taken from the National Demographic and Health Survey (NDHS) of Nepal, the study population was limited to government employees in Doti district. Therefore, it was both methodologically sound and necessary to use the finite population correction (FPC) based on the actual population of interest — i.e., the total number of government employees in Doti (N = 625).

Margin of Error Clarification:

By applying the finite population correction, the effective margin of error is slightly reduced compared to the initial 5% assumption, leading to more precise estimates.

2. Comment: Review the limitations of the Hosmer-Lemeshow test and interpret large confidence intervals for AOR (age group). Consider including additional model fit parameters, such as AIC or BIC.

Response: Thankyou for your thoughtful comments. We acknowledge and have addressed the concerns regarding model fit evaluation and the interpretation of regression outputs.

Limitations of the Hosmer-Lemeshow Test: While applying the multivariate analysis, the Hosmer and Lemeshow test was performed to test the goodness of fit. The test result (p = 0.345) indicated good model fit. However, we recognize that the Hosmer-Lemeshow test has limitations (line number 291-294). It is sensitive to sample size and can yield misleading results in small samples like ours (n = 195), where it may fail to detect misfit. Furthermore, the grouping of predicted probabilities into deciles is arbitrary and may affect the outcome. Therefore, we agree that relying solely on this test may not provide a full picture of model adequacy.

Interpretation of Wide Confidence Intervals (AORs): As observed in Table 3, some adjusted odds ratios (AORs), particularly for age groups 30–40 (AOR: 14.4; 95% CI: 1.6–127.7) and 40–50 (AOR: 13.7; 95% CI: 1.04–180.3), have very wide confidence intervals, suggesting a high degree of uncertainty around the estimates. This is likely due to small subgroup sizes or sparse data, which can inflate standard errors and widen confidence intervals. Despite statistical significance, these estimates should be interpreted with caution, as the precision is limited. This limitation has now been explicitly mentioned in the discussion section. (line number 243-246)

3. Comment: Correct the misinterpretation of non-significant results for factors such as male gender and physical activity (lines 239 and 254).

Response: We have revised these sections to reflect that the male gender and physical activity were not statistically significant risk factors.

4. Comment: Compare the significantly different hypertension prevalence rates (38% and 66%) between government employees in Nepal and other countries.

Response: We have clarified the comparison between the different prevalence rates and provided additional context regarding the behavioral and lifestyle differences that might account for this disparity.(line number 231-234)

We believe that these revisions have addressed the concerns of both reviewers that helped to improve quality of our paper. We appreciate your valuable feedback and hope that the revised manuscript is now suitable for publication.

Sincerely,

Sunita Sharma

On behalf of the co-authors

---

## [Decision Letter · Decision Letter 1]

26 May 2025

Dear Dr. Sharma,

Thank you for submitting your manuscript to PLOS ONE. After careful consideration, we feel that it has merit but does not fully meet PLOS ONE’s publication criteria as it currently stands. Therefore, we invite you to submit a revised version of the manuscript that addresses the points raised during the review process.

Dear Authors,

1) Kindly respond to the reviewers' comments and suggestions thoughtfully, considering various perspectives. This should encompass not only Q and A but also,

i) The research question is articulated effectively with supporting evidence.

ii) The most suitable methodology has been employed to derive the results.

iii) Your results are presented clearly, utilizing the best statistical analysis methods, and any discrepancies or errors should be noted, with mandetory consultation and verification from statisticians as needed. If you find that you cannot adequately address the reviewer's comments, please explain why.

iv) Ensure that your revision addresses all comments, is written in flawless English, presents tables and figures on the same pages without fragmentation, and has captions and notes placed appropriately.

2) Please include a few additional sentences in the second paragraph of the 'Introduction' addressing, "Why did you select government employees for the hypertension study, or what makes them at risk?

Good luck!

We look forward to receiving your revised manuscript.

Kind regards,

Chhabi Lal Ranabhat

Academic Editor

PLOS ONE

Reviewers' comments:

Reviewer's Responses to Questions

**Comments to the Author**

Reviewer #1: All comments have been addressed

Reviewer #2: (No Response)

2. Is the manuscript technically sound, and do the data support the conclusions?

Reviewer #1: Partly

Reviewer #2: Partly

3. Has the statistical analysis been performed appropriately and rigorously?

Reviewer #1: Yes

Reviewer #2: No

4. Have the authors made all data underlying the findings in their manuscript fully available?

Reviewer #1: Yes

Reviewer #2: Yes

5. Is the manuscript presented in an intelligible fashion and written in standard English?

Reviewer #1: Yes

Reviewer #2: Yes

Reviewer #1: Abstract: line no 7, "which are the leading causes" could be revised into" which remains the leading causes of morbidity and mortality" (or you can revise on your own preferences)

In the methodology sections of the abstract, please add "structured questionnaire adopted from the WHO STEPS survey tool was used" would improve clarity for the reader.

Introduction

ln line 52 , Author has started with Cardiovascular disease (CVD) , it would be more effective to begin with hypertension itself, since it is the central focus of the study and then connect it to its role as a leading modifiable risk factor for CVDs.

Please include a brief rationale at the end of the introduction explaining why government employees in a rural district like Doti were selected as the study population.

Methodology

In line no 93 , You have added the term " Miscellaneous service " please specify what is meant by miscellaneous service ?

Lines 94–95: You have used "were not included." Please revise this to: "Currently employed government staff who were present during the data collection period were included in the study, while those who were on leave or absent were excluded."

Line 134 mentions the Omron automatic BP monitor, but line 141 refers to the random zero sphygmomanometer. These are two different instruments. Please clarify which was actually used.

Results

Line 181: Replace "+_SD" with "± SD."

In the table, please correct the spelling of “Brahmin.”

It is recommended to round the percentages in the education and service type categories so they sum exactly to 100%, for clarity and consistency.

Line 211: The finding that alcohol consumption appears to be a protective factor against hypertension (AOR < 1) is unusual and not supported by most studies. This may be due to a small number of drinkers in the study or information bias. It is recommended that the authors clearly interpret this result in the results section and acknowledge it as a potential limitation in the discussion.

Discussion

In the discussion section, it is suggested to restructure the comparison flow. The authors currently begin by comparing their findings with global and LMIC data, followed by Nepal-specific studies. Please revise the discussion flow to start with Nepalese studies, then South Asian/regional, and finally global data for better context.

Please add a reference to support the conclusion in lines 277–281 regarding interventions and prevention.

Limitation:

Please paraphrase lines 289 and 290 for better readability and flow.

Reviewer #2: Dear Author,

Thank you all for the responses. However, I still did not find any satisfactory responses or corrections for the following comments:

1. Re-analysis with the corrected Margin of error for precision somewhere in result section, interpret and align with discussion.

2. As you have rebutted about the H-L test of goodness of fit, and also about the AIC and BIC as model parameters, how do you assure the fitness?

I hope you will address these issues this time.

With regards,

Reviewer

**Do you want your identity to be public for this peer review?** For information about this choice, including consent withdrawal, please see our Privacy Policy

Reviewer #1: No

Reviewer #2: No

---

## [Author Response · Author response to Decision Letter 2]

7 Jul 2025

Respected Chhabi Lal Ranabhat,

Academic Editor

PLOS ONE

I am writing in response to the comments provided by the reviewers regarding our manuscript PONE-D-25-05146R1 entitled "Prevalence of hypertension and its associated factors among government employees in Doti district of Nepal". We appreciate the time and effort invested by the reviewers and the editorial team in evaluating our work. After careful consideration of the reviewers' comments, we have addressed each point raised and provided our responses below:

Reviewer 1

Abstract

Comment: Line 7, “which are the leading causes,” could be revised.

Response: Revised to: “which remain the leading causes of morbidity and mortality globally.” [page no 2, line no26]

Comment: Methodology – Add clarity about the tool used.

Response: Added the sentence: “The structured questionnaire adopted from the WHO STEPS survey tool was used for data collection.” [page no 2, line no 35-36]

Introduction

Comment: Start with hypertension, not CVD, in line 52.

Response: Revised the paragraph to begin with hypertension and subsequently linked it to CVD as a modifiable risk factor as below. [page no 3, line no 53-60]

Hypertension is a leading modifiable risk factor for cardiovascular diseases (CVDs) and premature mortality worldwide. Between 1990 and 2019, global deaths from CVDs increased steadily from 12.1 million to 18.6 million respectively [1]. The prevalence of hypertension among adults is higher in Low Middle Income Countries (LMICs) (31.5%, 1.04 billion people) than in High-Income Countries (28.5%, 349 million people) [2]. The rising prevalence of hypertension, coupled with low medication adherence, poor blood pressure control, and limited knowledge about the condition, presents significant challenges, particularly in low- and middle-income countries (LMICs) [2].

Comment: Add rationale for selecting government employees in rural Doti.

Response: Included rationale at the end of the introduction as below. [page no 3, line no 84-88]

Doti district, one of the underprivileged rural and hilly areas of Nepal has insufficient health data, especially when it comes to non-communicable diseases like hypertension. Studying prevalence of hypertension among government employees in a rural setting like Doti district not only closes a significant evidence gap but also contributes to the development of targeted strategies for the prevention and management of hypertension in underprivileged areas of Nepal.

Methodology

Comment: Line 93 – Clarify "Miscellaneous service."

Response: Revised to specify: “Miscellaneous service includes other gazetted-level government employees falling outside administration or health like audit, foreign affairs, statistics, engineering, agricultural, and other services” [page no 4, line no 102-104]

Comment: Lines 94–95 – Revise inclusion/exclusion phrasing.

Response: Revised to: “Currently employed government staff who were present during the data collection period were included in the study, while those who were on leave or absent were excluded.” [page no 4, line no 104-106]

Comment: Clarify discrepancy between Omron monitor and random zero sphygmomanometer.

Response: Clarified in the manuscript: “Blood pressure was measured using an Omron automatic BP monitor and no random zero sphygmomanometer was used” [page no 6, line no 146-147]

Results

Comment: Line 181 – Replace "+_SD" with "± SD."

Response: Corrected to “± SD.” [page no 7, line no 193]

Comment: Correct spelling of “Brahmin” in the table.

Response: Spelling has been corrected. [page no 8, line no 200]

Comment: Round percentages to sum to 100%.

Response: Percentages have been rounded appropriately to ensure they sum to 100% in the education and service-type categories. [page no 8, line no 200]

Comment: Alcohol consumption appears protective; interpret and note as limitation.

Response: We have noted the unusual finding, and acknowledged it in the discussion as a potential limitation as, [page no 12, line no 263-269] and in the limitations [page no 14, line no 324-326]

Interestingly, alcohol consumption in our study showed an inverse association (AOR :0.35; 95% CI: 0.17-0.72), potentially reflecting reverse causation, reporting bias, or unmeasured confounding. It is possible that this observed association is influenced by residual confounding or reverse causation, whereby individuals with existing health conditions may reduce or abstain from alcohol use. Moreover, potential underreporting or misclassification of alcohol intake cannot be ruled out. This is a notable limitation of the study and underscores the need for longitudinal research to further explore this relationship.

Discussion

Comment: Restructure comparison flow from global to local.

Response: Revised the discussion structure flow from global to local context. [page no 11-13, line no 238-296]

Comment: Add reference to support intervention/prevention statements (lines 277–281).

Response: A supportive interventions are added and proper reference has been added as below [page no 13, line no 296].

Interventions tailored to workplace environments such as encompassing routine screenings, healthy diet and physical activity promotion, and alcohol education [22] are critically needed to address this emerging public health challenge.

Limitation

Comment: Paraphrase lines 289–290 for better clarity.

Response: Revised to: “The study adopted a cross-sectional design, which may preclude the establishment of causality with the identified risk factors. [page no 13 , line no 299-300]

Reviewer 2

Comment 1: Re-analysis with corrected margin of error for precision.

Response: We acknowledge the oversight. The margin of error has been corrected in the statistical analysis. The updated confidence intervals and interpretations are reflected in the results section and aligned appropriately in the discussion as: (page no 8-9, line no 201-210)

The prevalence of hypertension among government employee was found to be 36.4% ± 5.6% (36.4% ± 5.6%) (95% confidence, with finite population correction) as presented in Table 2. The margin of error for sample size (n=195) and total number of government employees in Doti (N = 625) with 95% Confidence Interval has been calculated as follows,

Step 1: Standard Error (without FPC)

SE= √((p(1-p))/n)= √((0.364(1-0.364))/195)= √0.0011872≈0.03447 Step 2: Finite Population Correction (FPC)

FPC= √((N-n)/(N-1))= √((625-195)/624)= √(430/624)= √({0.688} )≈0.8294________________________________________

Step 3: Corrected Margin of Error (MoE)

MoE=z×SE×FPC=1.96×0.03447×0.8294=1.96×0.02859≈0.05604

Margin of Error= 5.6%

Prevalence of hypertension=36.4% ± 5.60% (95% confidence, with finite population correction)

Comment 2: Clarify fitness of model in absence of H-L test, AIC, or BIC.

Response: We appreciate the reviewer’s valuable suggestion regarding additional metrics for model fit assessment. In our analysis, we used the Hosmer–Lemeshow (H–L) test to evaluate the goodness of fit for both models. The first model yielded a p-value of 0.345, and the second model had a p-value of 0.820—both exceeding the conventional threshold of 0.05, indicating acceptable model fit and no significant difference between observed and predicted probabilities.

In addition, we reported the Nagelkerke R² values, which were 0.445 and 0.091, respectively. These values suggest that the explanatory variables in the first model accounted for approximately 45% of the variation in the dependent variable, which represents a moderate explanatory power in epidemiological modeling. While the R² for the second model was relatively low, such values are typical in cross-sectional studies where numerous unmeasured or behavioral factors may influence outcomes.

Regarding AIC or BIC, we acknowledge their usefulness in model comparison and selection. However, as our analysis platform did not provide log-likelihood outputs, calculation of AIC/BIC was not feasible. But we believe that the combination of the Hosmer–Lemeshow test and Nagelkerke R² provides a standard and acceptable evaluation of model fit in cross-sectional logistic regression, particularly within resource-constrained research settings. Further, we have also noted the limitations of these methods in the revised manuscript.

We believe that these revisions have satisfactorily addressed the reviewers' comments and significantly improved the quality of our manuscript. We sincerely appreciate your valuable feedback and hope that the revised manuscript meets the publication standard of your reputed journal. Thank you for your time and consideration.

Sincerely,

Sunita Sharma

On behalf of the co-authors

---

## [Decision Letter · Decision Letter 2]

6 Aug 2025

Prevalence of hypertension and its associated factors among government employees in Doti district of Nepal

PONE-D-25-05146R2

Dear Dr. Sharma,

We’re pleased to inform you that your manuscript has been judged scientifically suitable for publication and will be formally accepted for publication once it meets all outstanding technical requirements.

Kind regards,

Chhabi Lal Ranabhat

Academic Editor

PLOS ONE

Additional Editor Comments (optional):

Reviewers' comments:

Reviewer's Responses to Questions

**Comments to the Author**

Reviewer #1: All comments have been addressed

Reviewer #2: All comments have been addressed

2. Is the manuscript technically sound, and do the data support the conclusions?

Reviewer #1: Yes

Reviewer #2: Yes

3. Has the statistical analysis been performed appropriately and rigorously?

Reviewer #1: Yes

Reviewer #2: Yes

4. Have the authors made all data underlying the findings in their manuscript fully available?

Reviewer #1: Yes

Reviewer #2: Yes

5. Is the manuscript presented in an intelligible fashion and written in standard English?

Reviewer #1: Yes

Reviewer #2: Yes

Reviewer #1: The authors have addressed all the previous comments and made the necessory revisions accordingly. This study adds important evidence regarding hypertension among government employees and is a valuable contributions.

Reviewer #2: Thank you for addressing the comments; now I am recommending further procedures for acceptance and publication.

Regards

Reviewer

**Do you want your identity to be public for this peer review?** For information about this choice, including consent withdrawal, please see our Privacy Policy

Reviewer #1: No

Reviewer #2: **Yes: ** Chiranjivi Adhikari

---

## [Editor Report · Acceptance letter]

PONE-D-25-05146R2

PLOS ONE

Dear Dr. Sharma,

I'm pleased to inform you that your manuscript has been deemed suitable for publication in PLOS ONE. Congratulations! Your manuscript is now being handed over to our production team.

Kind regards,

on behalf of

Dr. Chhabi Lal Ranabhat

Academic Editor

PLOS ONE